# Open and Closed Triple Inhaler Therapy in Patients with Uncontrolled Asthma

Serafeim-Chrysovalantis Kotoulas [1], Ioanna Tsiouprou [2], Kalliopi Domvri [3], Polyxeni Ntontsi [4], Athanasia Pataka [5,*,†] and Konstantinos Porpodis [2,†]

1   ICU, General Hospital of Thessaloniki "Hippokration", 49th Kostantinoupoleos Street, 57642 Thessaloniki, Greece; akiskotoulas@hotmail.com
2   Pulmonary Clinic of Aristotle University of Thessaloniki, General Hospital of Thessaloniki "Georgios Papanikolaou", Leoforos Papanikolaou, Exohi, 57010 Thessaloniki, Greece; joanna_tsi@hotmail.com (I.T.); kporpodis@yahoo.gr (K.P.)
3   Laboratory of Histology-Embryology, Medical School, Aristotle University of Thessaloniki, Leoforos Agiou Dimitriou, 54124 Thessaloniki, Greece; kdomvrid@auth.gr
4   2nd University Department of Respiratory Medicine, Attikon Hospital, 1st Rimini Street, Haidari, 12462 Athens, Greece; xenia-1990@hotmail.com
5   Respiratory Failure Clinic of Aristotle University of Thessaloniki, General Hospital of Thessaloniki "Georgios Papanikolaou", Leoforos Papanikolaou, Exohi, 57010 Thessaloniki, Greece
*   Correspondence: patakath@yahoo.gr; Tel.: +30-2313307178
†   These authors contributed equally to this work.

**Highlights:**

**What are the main findings?**

- The addition of a LAMA, for asthma symptoms and exacerbation control, in patients with persistent asthma, uncontrolled with medium or high dose ICS + LABA, is an effective treatmen option;
- Multiple inhaler devices could be one of the reasons for suboptimal adherence because it is challenging for the patients to establish and sustain the correct technique for each inhaler.

**What is the implication of the main finding?**

- LAMAs should be used as an add-on treatment for the control of symptoms and exacerbations in patients with asthma that remains persistently uncontrolled despite treatment with ICS + LABA;
- The use of a singletriple inhaler simplifies the treatment in contrast to an open triple inhaler, and this fact could strengthen adherence.

**Abstract:** Long-acting muscarinic antagonists (LAMAs) are a class of inhalers that has recently been included as add-on therapy in the GINA guidelines, either in a single inhaler device with inhaled corticosteroids plus long-acting β2-agonists (ICS + LABA) (closed triple inhaler therapy) or in a separate one (open triple inhaler therapy). This review summarizes the existing evidence on the addition of LAMAs in patients with persistently uncontrolled asthma despite ICS + LABA treatment based on clinical efficacy in the reduction of asthma symptoms and exacerbations, the improvement in lung function, and its safety profile.

**Keywords:** asthma; ICS + LABA; LAMA; GINA; closed triple inhaler therapy; review; biomarker-guided treatment

## 1. Introduction

Inhaled corticosteroids (ICS) are the cornerstone of asthma treatment [1,2]. Along with long-acting β2-agonists (LABAs), they compose the main treatment option of "steps 3–5", according to Global Initiative for Asthma (GINA) guidelines, in patients with moderate to severe asthma [3,4]. Patients who are unable to maintain satisfactory asthma control or experience frequent exacerbations, despite good adherence to treatment with ICS + LABA

and adequate management of comorbidities, may benefit from new treatment options [5]. Monoclonal antibodies as phenotype-guided treatment optionsand a well-established drug class, namely the long-acting muscarinic antagonists (LAMAs), are included in these breakthrough therapeutic strategies [6,7]. Monoclonal antibodies have been approved as an option for severe asthma management at "step 5" of GINA guidelines in both adults and children [8,9]. LAMAs are considered as an add-on therapy at "steps 4 and 5" of GINA guidelines, either in a single inhaler device combined with ICS and LABA (closed triple inhaler therapy, only for adults) or in a separate inhaler device (open triple inhaler therapy, for both children and adults), enriching, in this way, the physicians' arsenal against the disease [3,10,11].

Although LAMAs are one of the main options for the treatment of Chronic Obstructive Pulmonary Disease (COPD), until recently, there was no supporting evidence for use in the management of asthmatic patients in everyday clinical practice [9]. As far as anticholinergics, only short-acting muscarinic antagonists (SAMAs) were used during asthma exacerbations [12,13] as one of numerous emergency treatment pharmacological strategies, such as ketamine, aminophylline, and fentanyl [14–16]. LAMAs were recently studied for the management of asthma; despite the different pathogenetic mechanisms between asthma and COPD, based on the common cardinal pathophysiological characteristic of bronchoconstriction [17,18], the bronchodilator action of LAMAs [19] and the effectiveness of SAMAs in asthma exacerbation treatment [20]. Furthermore, a few trials focused on identifying patients that could specifically benefit from LAMA treatment initiation based on disease traits and various biomarkers [21].

The purpose of this review is to consolidate current knowledge on the effect of LAMAs—mainly glycopyrrolate bromide and tiotropium bromide, which demonstrate favorable pharmacological and safety properties—on asthma control, asthma exacerbation rate, and lung function [9]. The authors also investigate the role of several biomarkers in selecting patients for treatment with LAMAs, as well as LAMAs' safety profile on asthmatic populations.

## 2. Discussion

### 2.1. Asthma Control

The assessment of asthma control is based on the use of various questionnaires;the most widely used are the St. George respiratory questionnaire (SGRQ) [22], the asthma control questionnaire (ACQ) [23], the asthma quality of life questionnaire (AQLQ) [24], and the asthma control test (ACT) [25]. The total score of each test is calculated according to the patient'sanswers, and subsequently, the level of asthma control is classified [26–29].

A retrospective observational study demonstrated that severe asthmatics under treatment with LAMAs used to experience significantly worse asthma control and worse disease-related quality of life before LAMA initiation, based on the results of ACT, ACQ, and AQLQ [10]. These patients tended to be ex-smokers and had later-onset asthma in comparison to patients who did not receive LAMAs. Moreover, they were often diagnosed with concomitant bronchiectasis and used to receive oral corticosteroid (OCS) treatment courses and long-term treatment with monoclonal antibodies more frequently compared with non-LAMAusers [10].

In the TRIGGER study, a randomized controlled trial (RCT) on patients with uncontrolled asthma, which compared standard high-dose ICS + LABA treatment to open and closed triple inhaler therapy (ICS + LABA + LAMA), participants on close triple therapy experienced considerable improvement in their ACQ score and in the proportion of days free of asthma symptoms, at 52 weeks, but not in rescue medication use [30]. However, in the TRIMARAN study, a twin study with TRIGGER but with a medium ICS dose, no significant differences between the study groups in any of the aforementioned parameters were observed [30]. Similarly, no differences were found in the TRIGGER study between open and closed triple inhaler therapy arms [30]. In another randomized, non-inferiority study, the ARGON study, single-triple inhaler therapy was non-inferior to open triple

inhaler use as far as the AQLQ improvement at 24 weeks is concerned. The single triple inhaler therapy was superior in the ACQ and SGRQ improvement compared to open triple inhaler therapy in the ARGON study [31].

In the CAPTAIN study, an RCT like TRIMARAN, with a study period of 24 weeks, showed no clinically important improvement in SGRQ between study arms [32]. However, the minimal clinically important improvement in the ACQ was significantly greater regarding closed triple inhaler therapy, but this effect was only evident in the higher LAMA dosage study arm [32]. In the IRIDIUM study, another similar 52-week RCT, improvement in the ACQ questionnaire was significantly greater in the closed triple inhaler therapy group compared to the group that received only ICS + LABA therapy [33].

As far as asthma COPD overlap (ACO) is concerned, two studies showed no statistically significant difference in asthma control, measured with ACT, when a LAMA was added to the treatment with ICS + LABA [34,35]. It is noteworthy that the choice of rescue medication is different in patients receiving single-triple versus open-triple treatment. According to GINA guidelines, patients in single triple or open triple with no formoterol-containing regimen should use SABA as a reliever therapy and move to track 2 [3]. Nevertheless, there is no adequate evidence to advise in favor of using SABA combined with ICS (same or separate device) in these therapeutic categories [3].

Summarizing the existing evidence, it seems plausible to suggest the addition of a LAMA, for asthma symptom control, in patients with persistent asthma, uncontrolled with medium or high dose ICS + LABA treatment, either in a separate or the same inhaler device, which is more cost-effective [36]. Also, the use of a single triple inhaler simplifies the treatment in contrast to an open triple inhaler, and that fact could strengthen adherence. Multiple inhaler devices could be one of the reasons for suboptimal adherence because it is challenging for the patients to establish and sustain the correct technique for each inhaler [37].

*2.2. Asthma Exacerbation Rate*

The prevention of exacerbations, along with symptom control, are the two main indices of asthma management. Even patients with mild asthma or well-controlled asthma symptoms could experience severe exacerbations [38,39], which could be fatal [40–43].

The study byPuggioni et al. showed that asthmatic patients, who are already on LAMA treatment, have a significantly higher annual exacerbation rate [10]. In another retrospective observational study, Averell et al. demonstrated that asthma exacerbations were more frequent in patients receiving various control medications, including LAMAs [44]. Another study from Japan also showed that patients on triple inhaler therapy experienced an exacerbation in the year prior to triple treatment initiation significantly more frequently than those on ICS + LABA alone. However, this frequency was significantly lower in the year following the triple treatment initiation [45].

The patients under triple inhaler treatment (open or closed) in the TRIMARAN and TRIGGER studies demonstrated a significant reduction in the rate of moderate and severe exacerbations compared to those on ICS + LABA alone, by 15% and 12%, respectively, with no significant differences between the open and closed treatment combinations [30]. These reductions, however, showed a significant seasonal variation, as they were greater in winter, when the reduction in the rate of moderate and severe exacerbations with triple inhaler treatment was 20.3% compared to ICS + LABA alone, while in the other seasons of the year the corresponding reductions varied between 8.6% and 12.0% [46]. Furthermore, patients with higher reversibility in post-bronchodilator forced expiratory volume in 1 s (post-FEV$_1$) (>400 mL) experienced greater exacerbation reduction benefit. Nevertheless, no differences were observed between patients with different blood eosinophil levels [47].

In two older RCTs, the addition of a LAMA to ICS + LABA increased the time to the first severe exacerbation by 56 days and significantly reduced the risk for a severe exacerbation by 21% [48]. In the IRIDIUM study, the addition of a LAMA reduced the overall risk for an exacerbation along with the risk for moderate and/or severe exacerbation. Interestingly,

this reduction was more evident in high ICS dosages [33], while in the CAPTAIN study, only non-significant reductions were observed in moderate and/or severe exacerbation rates. Similarly, those reductions were more evident in the study arms of higher ICS doses [32]. Regarding ACO, there is no clear evidence whether ICS + LABA + LAMA treatment reduces the time to first exacerbation [35].

As a result, the addition of a LAMA to the standard treatment with ICS + LABA probably reduces the risk of moderate and/or severe asthma exacerbations and should be used in cases wheredual treatment is inadequate in the prevention of the asthma exacerbations [21,36].

### 2.3. Lung Function

Lung function is another key aspect of asthma, as it might mirror disease control. In contrast to asthma symptoms and exacerbation control, the addition of a LAMA to the dual therapy of ICS + LABA in patients with asthma that remains uncontrolled leads to a clearly beneficial effect on lung function [46]. For this reason, patients on LAMA + ICS + LABA usually presented with worse lung function during the initial clinical assessment [10].

In the CAPTAIN study, the addition of LAMA to ICS + LABA increased the $FEV_1$, between 82 and 110 mL in the different study arms, with different drug doses—and these differences were statistically significant across all study arms and supported by the analysis of $FEV_1$ at 3 h post-dose [42]. Similar results were also seen in the IRIDIUM study, in which the statistically significant increase in trough $FEV_1$ varied between 65 and 119 mL [33]. In two older RCTs by Kerstjens et al., both pre-dose (trough) and peak $FEV_1$ were significantly increased after LAMA initiation [48].

In the two parallel-group RCTs, the TRIMARAN and TRIGGER studies, the participants in the LAMA study arms experienced significant improvement in both pre-dose and peak $FEV_1$, as well as in morning peak expiratory flow (PEF) [30]; moreover, in the TRIMARAN study, the sub-group of patients with low eosinophil blood count (<300 cells/μL), experienced greater benefit, although this was not evident in the TRIGGER study [47]. The latter also showed no differences between the open and the closed triple inhaler therapy study arms [30]. Nevertheless, in the ARGON study, closed triple inhaler therapy with a high ICS dose was superior to the open triple inhaler therapy regarding the improvement of trough $FEV_1$ and of morning and evening PEF. The corresponding results between the medium ICS dose closed triple inhaler therapy and open triple inhaler therapy were comparable [31].

The addition of LAMA to the ICS + LABA regimen seems to improve lung function in patients with ACO. This fact is supported by the results of two studies that showed a significant improvement in multiple pulmonary function tests (PFTs) and impulse oscillometry (IOS) parameters after LAMA treatment initiation [34,35].

To sum up, the addition of LAMA to the standard ICS + LABA treatment regimen undoubtedly significantly improves lung function in patients with inadequately controlled asthma [21,36].

### 2.4. Asthma-Related Biomarkers

There is a great challenge in deciding the next step of treatment in patients with severe, uncontrolled asthma between the choices to addon LAMA on ICS + LABA treatment or to increase the ICS dose in the ICS + LABA regimen. In a retrospective cohort study, the addition of LAMA to ICS + LABA decreased the risk of exacerbations by 35% versus ICS dose escalation [49]. As discussed in previous sections, the addition of LAMA improved lung function and daily symptom control [50]. It would be helpful to have asthma biomarkers that could detect patients who could benefit the most from LAMA add-on therapy.

Data from national registries of patients with severe asthma have shown that patients on LAMA treatment have higher eosinophil blood count levels. However, a fraction of exhaled nitric oxide (FeNO) and serum total IgE levels are comparable with those of non-LAMAusers [10]. There is a limited number of studies thatinvestigate the eligible profile of

severe, refractory asthma patients for LAMA initiation. As mentioned above, in the TRI-MARAN study, the sub-group of patients with low eosinophil blood count (<300 cells/µL) experienced greater improvement in lung function after LAMA initiation [47]. This was not the case in the TRIGGER study. In this study, eosinophil blood count was not associated with the rate of exacerbations [47]. Moreover, in the CAPTAIN study, clinic trough $FEV_1$ and annual moderate and/or severe exacerbation rates were not affected by LAMA initiation. They were affected by ICS dosage in the patient groups with higher baseline FeNO and blood eosinophil count [32]. Furthermore, the beneficial effect of LAMA in patients with ACO seems to be independent of FeNO level, total serumIgE, or eosinophil blood count [34].

Currently, there are no reliable biomarkers that could identify patients with severe refractory asthma eligible for LAMA treatment initiation. Some asthma traits that might be predictive ofbetter response to treatment with LAMA could be a previous smoking history and fixed airway obstruction [36]. The role of other recognizable disease traits is still under investigation [21].

*2.5. Safety*

There have been several concerns about LAMAs' safety, especially due to the cardiovascular adverse effects following their anticholinergic action. However, several studies have shown that in asthmatic patients, they demonstrate a particularly favorable safety profile.

In the CAPTAIN study, adverse events, as well as serious adverse events, were comparable among study groups, with only one death to be considered as related toa study drug in the study arm of the triple therapy with the low concentration of umeclidinium [32]. The same trend in adverse events was also reported in the IRIDIUM study, in which seven deaths were reported, which were balanced between LAMA and non-LAMA treatment arms, and none of them was considered, by the investigators, to be related toany of the study drugs [33]. Similar safety outcomes were also reported in the TRIMARAN and TRIGGER studies [30], while Kerstjens et al. reported no deaths in their RCT a few years earlier [48]. In addition, in the ARGON study, mild adverse events were comparable between treatment groups. The most frequent serious adverse event was pneumonia, which was correlated with the increased dose of ICS rather than the addition of LAMAs. Also, the only death that occurred was caused by a hemorrhagic stroke and was not considered to be study drug-related [31].

Consequently, existing evidence suggests that the safety profile of adding LAMA to the standard treatment with ICS + LABA is excellent [36]. Table 1 presents the main characteristics of all studies included in this review.

**Table 1.** Main characteristics of the studies included in this review.

| Reference | Study Type | Participants | Main Findings | Main Limitations |
|---|---|---|---|---|
| Puggioni 2020 [10] | Retrospective observational study. | 698 patients from Severe Asthma Network in Italy (SANI). | Severe asthmatics under treatment with long-acting muscarinic antagonists (LAMAs) experience significantly worse asthma control and worse disease-related quality of life before LAMA initiation. Asthmatic patients, who are already on LAMA treatment, have a significantly higher annual exacerbation rate. In LAMAusers, the fraction of exhaled nitric oxide (FeNO) and serum total immunoglobin E (IgE) levels were comparable with those of non-LAMAusers. | These patients tended to be ex-smokers and to have later-onset asthma in comparison to patients who did not receive LAMAs. They were also more often diagnosed with concomitant bronchiectasis and used to receive oral corticosteroid (OCS) treatment courses and long-term treatment with monoclonal antibodies more frequently compared to non-LAMAusers. |
| Virchow 2019 [30] | Two randomized controlled trials (TRIMARAN & TRIGGER). | 1155 asthma patients in TRIMARAN & 1437 in TRIGGER. | In the TRIGGER study, participants on close triple therapy experienced considerable improvement in their asthma control questionnaire (ACQ) score and in the proportion of days free of asthma symptoms at 52 weeks, but not in rescue medication use. In the TRIMARAN study, no significant differences between the study groups in any of the aforementioned parameters were observed. In both studies, a significant reduction in the rate of moderate and severe exacerbations was observed. In both studies, a significant improvement in both pre-dose and peak forced expiratory volume in 1 s ($FEV_1$), as well as in morning peak expiratory flow (PEF), was observed. In both studies, adverse events, as well as serious adverse events, were comparable among the study groups. | No differences were found in the TRIGGER study, as far as asthma control, between open and closed triple inhaler therapy arms. No significant differences between the open and closed treatment combinations in asthma exacerbation rate control were found in either study. No significant differences between the open and the closed triple inhaler therapy study arms in lung function were found in either study. |
| Gessner 2020 [31] | Randomized, non-inferiority study (ARGON). | 1426 patients with asthma. | Singletriple inhaler therapy was non-inferior to open triple inhaler use as far as the asthma quality of life questionnaire (AQLQ) improvement at 24 weeks is concerned. The single triple inhaler therapy proved superior in the ACQ and St. George respiratory questionnaire (SGRQ) improvement. Closed triple inhaler therapy with a high ICS dose was superior to the open triple inhaler therapy regarding the improvement of trough $FEV_1$ and of morning and evening PEF. Adverse events were comparable among study groups. | Partially blinded design. Relatively short duration of the trial for evaluation of exacerbations (24 weeks). No significant differences between the studied groups in asthma exacerbation rate control were observed. While current smokers were included, they accounted for a minor subgroup (2.2%);thus, no firm conclusions can be drawn regarding the effect of current smoking and its potential influence on the study outcomes. |
| Lee 2021 [32] | Randomized controlled trial (CAPTAIN). | 2439 patients with asthma. | The minimal clinically important improvement in the ACQ was significantly greater regarding closed triple inhaler therapy. The addition of LAMA increased the $FEV_1$, between 82 and 110 mL in the different study arms, with different drug doses—and these differences were statistically significant across all study arms and were supported by the analysis of $FEV_1$ at 3 h post-dose. | There was no clinically important improvement in the SGRQ between study arms. In the patient groups with higher baseline FeNO and blood eosinophil count, clinic trough $FEV_1$ and annual moderate and/or severe exacerbation rate were not affected by LAMA initiation but by inhaled corticosteroids (ICS) dosage. |

**Table 1.** *Cont.*

| Reference | Study Type | Participants | Main Findings | Main Limitations |
|---|---|---|---|---|
| Kerstjens 2020 [33] | Randomized controlled trial (IRIDIUM). | 3092 patients with asthma. | ACQ questionnaire improvement was significantly greater in the closed triple inhaler therapy group compared to the group that received only ICS + long-acting $\beta_2$-agonists (LABA) therapy. The addition of a LAMA reduced the overall risk for an exacerbation along with the risk for moderate and/or severe exacerbation. The addition of LAMA to ICS + LABA increased the $FEV_1$ between 65 and 119 mL. Adverse events were comparable among study groups. | The reduction in the overall risk for an exacerbation and the risk for moderate and/or severe exacerbation was more evident in high ICS dosages. The study was not powered to provide conclusive answers for some comparisons and endpoints. This is a carefully controlled study and, therefore, not necessarily reflective of a real-world setting. |
| Ishiura 2019 [34] | Randomized, open-label crossover pilot study. | 17 asthma chronic obstructive pulmonary disease (COPD) overlap (ACO) patients. | No statistically significant difference in asthma control, measured with ACT, when a LAMA was added to the treatment with ICS + LABA. Significant improvement in multiple pulmonary function tests (PFTs) and impulse oscillometry (IOS) parameters after LAMA treatment initiation. The beneficial effect of LAMA in patients with ACO seems to be independent of FeNO level, total serumIgE, or eosinophil blood count. | The results by themselves cannot express the etiology of the clinically beneficial effect. A specific, formal definition of ACO has yet to be determined. The number of patients enrolled in this study was insufficient to detect benefits with respect to healthcare outcomes. This study was also relatively short for the assessment of patient-reported outcomes. |
| Park 2021 [35] | Multicenter, 48-week, randomized, non-inferiority trial (ATOMIC). | 303 patients with ACO. | No statistically significant difference in asthma control, measured with ACT, when a LAMA was added to the treatment with ICS + LABA. There is no clear evidence whether ICS + LABA + LAMA treatment reduces the time to first exacerbation. Significant improvement in multiple pulmonary function tests (PFTs) and impulse oscillometry (IOS) parameters after LAMA treatment initiation. | The study uses a non-inferiority study design. The incidence of adverse events was much less than expected. The definition of ACO is still controversial. This study was conducted as an open-label study, where subject bias can affect the result. |
| Braido 2022 [36] | Review. | Previously published studies. | It seems plausible to suggest the addition of a LAMA, for asthma symptom control, in patients with persistent asthma, uncontrolled with medium or high dose ICS + LABA treatment, either in a separate or the same inhaler device, which is more cost-effective. The addition of a LAMA to the standard treatment with ICS + LABA probably reduces the risk of moderate and/or severe asthma exacerbations and should be used in cases wheredual treatment is inadequate in the prevention of asthma exacerbations. The addition of LAMA to the standard ICS + LABA treatment regimen undoubtedly significantly improves lung function in patients with inadequately controlled asthma. Some asthma traits that might be predictive ofbetter response to treatment with LAMA could be a previous smoking history and fixed airway obstruction. Existing evidence suggests that the safety profile of adding LAMA to the standard treatment with ICS + LABA is excellent. | Available data related to the impact of triple inhaled therapy on asthma control and quality of life are conflicting. Currently, there are no reliable biomarkers that could identify patients with severe refractory asthma eligible for LAMA treatment initiation. |

**Table 1.** *Cont.*

| Reference | Study Type | Participants | Main Findings | Main Limitations |
|---|---|---|---|---|
| Averell 2019 [44] | Retrospective observational study. | 1821 patients treated with tiotropium. | Asthma exacerbations were more frequent in patients receiving various control medications, including LAMAs. | It was not possible to confirm that tiotropium was prescribed specifically for the treatment of asthma. Treatment patterns identified in this study may change over time. |
| Suzuki 2020 [45] | Retrospective, observational cohort study. | 1546 patients with asthma and 199 patients with ACO. | The exacerbation rate was significantly lower in the year following the triple treatment initiation compared to patients receiving ICS + LABA alone. | The exclusion of patients who withdrew from the health insurance database may have resulted in the exclusion of patients who died due to severe asthma. Approximate severity was estimated using ICS daily dose, meaning that severity could only be approximated for patients on a fixed daily ICS dose and not for patients on a variable ICS dose. |
| Papi 2022 [46] | Post-hoc analyses of TRIMARAN and TRIGGER | 1155 asthma patients in TRIMARAN and1437 in TRIGGER. | Significant reduction in the rate of moderate and severe exacerbations with seasonal variation, as they were greater in winter compared to the other seasons (20.3% and between 8.6% and 12.0%, respectively). The addition of a LAMA to the dual therapy of ICS + LABA in patients with asthma that remained uncontrolled led to a clearly beneficial effect on lung function. | These results came from post-hoc analyses. The conclusions were drawn on mean data rather than individual patient analyses and didnot provide a comprehensive mechanistic explanation for the results that we observed. |
| Singh 2020 [47] | Post-hoc analyses of TRIMARAN and TRIGGER. | 1155 asthma patients in TRIMARAN and1437 in TRIGGER. | Patients with higher reversibility in post-bronchodilator $FEV_1$ (>400 mL) experienced greater exacerbation reduction benefits. In the TRIMARAN study, the sub-group of patients with low eosinophil blood count (<300 cells/$\mu$L) experienced greater benefits in lung function. | No differences were observed between patients with different blood eosinophil levels in exacerbation rate reduction. In the TRIGGER study, eosinophil blood count was not associated with the lung function benefit. |
| Kerstjens 2012 [48] | Two replicate, randomized, controlled trials. | 912 patients with asthma. | The addition of a LAMA to ICS + LABA increased the time to the first severe exacerbation by 56 days and significantly reduced the risk for a severe exacerbation by 21%. Both pre-dose (trough) and peak $FEV_1$ were significantly increased after LAMA initiation. No deaths were reported. | Inconsistency in the results between the two trials. A larger placebo response was seen in trial 1 than in trial 2. The trials were supported by two pharmaceutical companies. |
| Cazzola 2022 [21] | Review. | Previously published studies. | The addition of a LAMA to the standard treatment with ICS + LABA probably reduces the risk of moderate and/or severe asthma exacerbations and should be used in cases wheredual treatment is inadequate in the prevention of asthma exacerbations. The addition of LAMA to the standard ICS + LABA treatment regimen undoubtedly significantly improves lung function in patients with inadequately controlled asthma. | The role of several recognizable disease traits in treatment decisions is still under investigation. The study was supported by a pharmaceutical company. |

**Table 1.** *Cont.*

| Reference | Study Type | Participants | Main Findings | Main Limitations |
|---|---|---|---|---|
| Chipps 2020 [49] | Retrospective cohort study. | 7857 patients with asthma. | The addition of LAMA to ICS + LABA decreased the risk of exacerbations by 35% versus ICS dose escalation. | Lack of information about events and rate of events that did not result in a paid claim. Actual SABA use, inhaler technique, medication adherence, and influence of comorbidities on uncontrolled symptoms. |
| Peters 2010 [50] | Three-way, double-blind, triple-dummy crossover trial. | 210 patients with asthma. | The addition of LAMA improved lung function and daily symptom control. | Evaluated only a small number of patients, with no treatment lasting longer than 14 weeks. Could not examine either the rate of asthma exacerbations or long-term safety issues. |

### 2.6. Single Maintenance and Reliever Therapy (SMART) vs. Single Triple Inhaler Therapy

Single maintenance and reliever therapy (SMART) is constituted by a single inhaler thatcontains ICS + formoterol [51,52]. As formoterol in inhaler devices exhibits both short periods of initiation of action and long duration of action, it can be used both as a reliever and maintenance therapy. These properties of formoterol have led GINA to recommend the maintenance and reliever therapy (MART) of budesonide or beclomethasone + formoterol taken as needed in steps 1 and 2 or regularly in steps 3, 4, and 5 as the preferred treatment regimen compared to ICS ± LABA (other than formoterol) as maintenance therapy + SABA as a reliever, in its latest guidelines [3].

In the same guidelines, as previously discussed, LAMAs were introduced as an add-on treatment in steps 4 and 5, either with a single or an additional inhaler device. In the case of the single triple inhaler therapy, there are three available options: (a) beclomethasone or budesonide + formoterol + glycopyrronium, (b) fluticasone furoate + vilanterol + umeclidinium, and (c) mometasone + indacaterol + glycopyrronium; it is clearly stated, although not supported by relevant studies yet, that patients who use the first option (beclomethasone or budesonide + formoterol + glycopyrronium), which contains formoterol, are able to continue using their inhaler for maintenance and as a reliever therapy [3].

Therefore, there is no actual dilemma between SMART and single triple inhaler therapy in the treatment steps 4 and 5, as the preferred choice would be the option of beclomethasone or budesonide + formoterol + glycopyrronium as a SMART with a triple inhaler or SMARTTI (Figure 1).

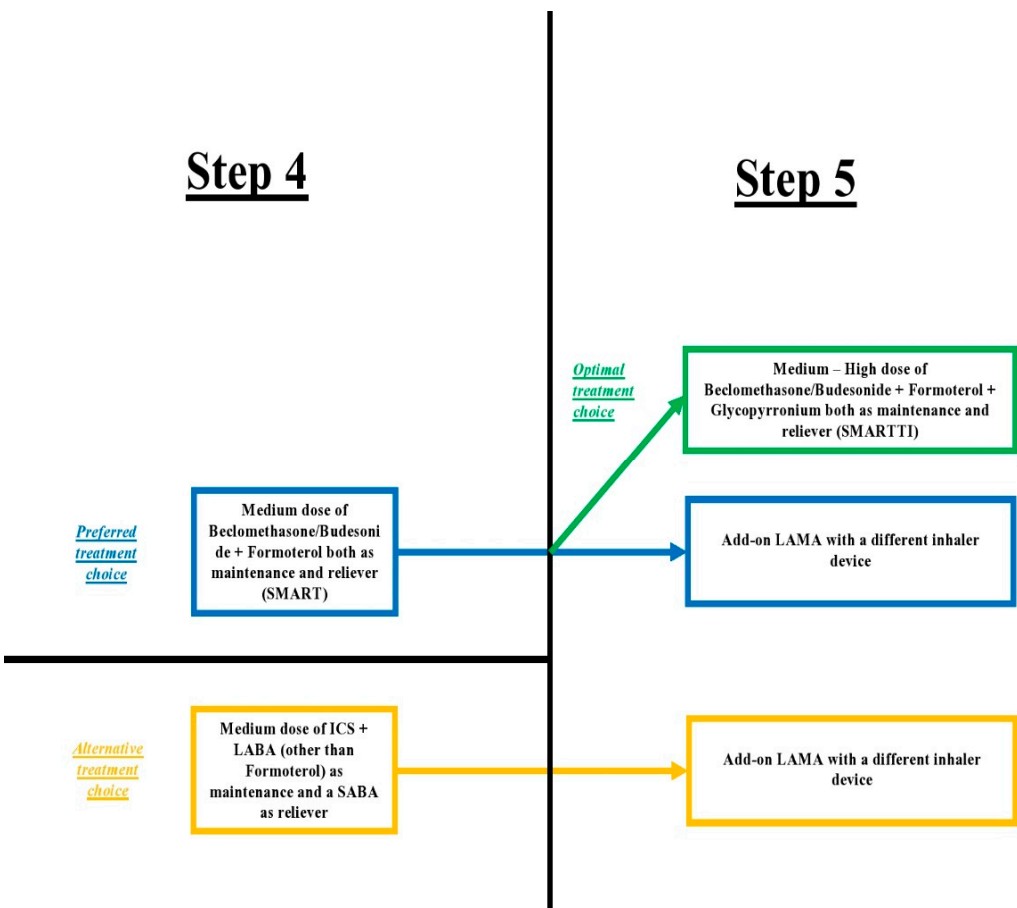

**Figure 1.** Optimal treatment choice in asthma patients when adding a LAMA to the combination of ICS + LABA.

## 3. Conclusions

In conclusion, LAMAs seem to present a reliable add-on treatment for the control of symptoms and exacerbations in patients with asthma that remains persistently uncontrolled despite treatment with ICS + LABA (Figure S1). Furthermore, current evidence shows that LAMAs improve lung function in this group of patients, and they present a satisfactory safety profile. Given the effectiveness and the safety profile of LAMAs in this group of patients, although it is not supported by any relevant studies, at least for now, this review is the first to propose the combination of the preferable "Single Maintenance And Reliever Therapy" (SMART) with the "Single Inhaler Triple Therapy" (SITT) in a single device as the "optimal treatment strategy" for these patients.

Consequently, there is a need forfurther investigation to detect the subgroup of severe asthma patients that would benefit the most from such a treatment. Nowadays, it remains unclear whether a singletriple inhaler therapy outmatches the open one in a variety of asthma-related factors other than cost-effectiveness.

**Supplementary Materials:** The following supporting information can be downloaded at: https://www.mdpi.com/article/10.3390/arm91040023/s1, Figure S1: Existing evidence for the addition of LAMAs in patients with persistently uncontrolled asthma despite ICS + LABA treatment

**Author Contributions:** Conceptualization, S.-C.K., A.P. and K.P.; methodology, S.-C.K. and I.T.; formal analysis, S.-C.K. and A.P.; investigation, K.D.; resources, I.T., K.D. and P.N.; data curation, P.N. and I.T.; writing—original draft preparation, S.-C.K., A.P., K.P. and I.T.; writing—review and editing, A.P. and K.P.; visualization, S.-C.K.; supervision, A.P. and K.P. All authors have read and agreed to the published version of the manuscript.

**Funding:** This research received no external funding.

**Conflicts of Interest:** The authors declare no conflict of interest.

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
