# Peer review of "Open and Closed Triple Inhaler Therapy in Patients with Uncontrolled Asthma"

_arm, doi:10.3390/arm91040023_

Round 1
Reviewer 1 Report
It was a great opportunity to revise this interesting review by Kotoulas et al. on the effectiveness of triple inhaler therapy in patients with uncontrolled asthma and its role in improving adherence to therapy. The review is sound and very well written. I have only a minor issue to be addressed.
Introduction
- Line 63-64. Not only short acting muscarinic antagonists (SAMAs) have been used, as an emergency treatment, during asthma exacerbations. In fact, authors should also mention other pharmacological strategies, such as ketamine (doi: 10.1007/s00228-022-03374-3) aminophylline (doi: 10.4103/1817-1737.191874) and fentanyl (doi: 10.5114/ait.2020.97765). Please modify and add these 3 references.
Author Response
Reply to reviewer 1
First of all, we would like to thank the reviewer for the much-appreciated time and effort dedicated to our submission as well as for the valuable recommendation that was suggested.
Reviewer
- Line 63-64. Not only short acting muscarinic antagonists (SAMAs) have been used, as an emergency treatment, during asthma exacerbations. In fact, authors should also mention other pharmacological strategies, such as ketamine (doi: 10.1007/s00228-022-03374-3) aminophylline (doi: 10.4103/1817-1737.191874) and fentanyl (doi: 10.5114/ait.2020.97765). Please modify and add these 3 references.
Response: Thank you for your valuable suggestion. In our revised manuscript, we have added these pharmacological strategies, along with the corresponding references.
Reviewer 2 Report
This well-written review evaluated LAMA addition in patients with persistently uncontrolled asthma despite ICS + LABA treatment on asthma control, exacerbations and lung function, and concluded that no reliable biomarkers currently identify patients that may benefit the most from triple therapy.
Some questions: Why was the review by Cazzola et al and Braido et al. included as studies in this review? Few other recent reviews are not reffered to, such as DOI: 10.1080/17476348.2023.2167715 , DOI: 10.4103/atm.atm_341_22 or DOI: 10.1080/02770903.2023.2188556 . Moreover, it would benefit this review to discuss which novelty it adds compared to all previous efforts.
That there is no evidence yet to prefer SMART with a single triple inhaler therapy may be better reflected in the conclusions which are now quite strong.
Please specify in the comment regarding seasonal variation (p4L147) to which comparison the 20.3%, 8.6% and 12.0% refer.
Regarding safety, please add (p5L231) in which arm the fatal case in the CAPTAIN study occurred.
Author Response
Reply to reviewer 2
First of all, we would like to thank the reviewer for the much-appreciated time and effort dedicated to our submission as well as for the valuable comments and recommendations that were suggested.
Reviewer
Some questions: Why was the review by Cazzola et al and Braido et al. included as studies in this review? Few other recent reviews are not referred to, such as DOI: 10.1080/17476348.2023.2167715 , DOI: 10.4103/atm.atm_341_22 or DOI: 10.1080/02770903.2023.2188556 . Moreover, it would benefit this review to discuss which novelty it adds compared to all previous efforts.
Response: Thank you for the comment. As far as for the studies that were included in this review, we tried to include, mainly, the large RCTs, of the last decade, about the addition of LAMAs to ICS + LABA treatment regimens, along with some other studies with the same research question.
We have chosen to include the two studies by Cazzola et al and Braido et al, but only referred to them in the concluding paragraph of each of the sections of our own review, in order to exhibit that other recently published reviews have also concluded in the same findings with us. As far as the other 3 studies that you mentioned, the 1st and the 3rd are so recent that had not been published when we were writing our own review, while the 2nd was not included in our review simply because we wanted to include in our review as few other reviews as possible. Nevertheless, we could include those 3 studies in our own, if you think that it would add merit to it.
As far as the novel findings of our own review compared to other previously published ones, it is true that there are not many novel things in our review compared with others. However, in the section 2.6 of our review, although it is not supported by any studies, at least for now, we propose the combination of the preferable “Single Maintenance And Reliever Therapy” (SMART) with the “Single Inhaler Triple Therapy” (SITT) in a single device as the “optimal treatment strategy” for these patients and we also quote a corresponding figure (figure 1) in which we illustrate this treatment strategy, so that it could be interpretented in an easier way. Nevertheless, since this novelty is not clearly discussed as such, we also added a corresponding sentence in the conclusion section of our revised manuscript, as you suggested.
Reviewer
That there is no evidence yet to prefer SMART with a single triple inhaler therapy may be better reflected in the conclusions which are now quite strong.
Response: Thank you for this valuable comment. We believe that with the addition of the corresponding sentence in the conclusion section of our revised manuscript, based on your previous comment, we also respond to the present comment as well.
Reviewer
Please specify in the comment regarding seasonal variation (p4L147) to which comparison the 20.3%, 8.6% and 12.0% refer.
Response: Thank you for bringing this to our attention. We specified it in the revised version of our manuscript. In winter, the reduction in the rate of moderate and severe exacerbations with triple inhaler treatment was 20.3% compared to ICS + LABA alone, while in the other seasons of the year the corresponding reductions varied between 8.6% and 12.0%.
Reviewer
Regarding safety, please add (p5L231) in which arm the fatal case in the CAPTAIN study occurred.
Response: Thank you for bringing this up. We added that in the revised version of our manuscript. It was in the study arm of the triple therapy with the low concentration of umeclidinium.
Reviewer 3 Report
This review add nothing or little to what is already available in the literature. It could be suitable as a letter after significant modifications.
Almost half of the references are old/very old (more than 5, 10, and even 20 years ago).
Section 2.6:
This change in the GINA guideline was not in 2022 as you cited, but in 2019 version of GINA.
I understand what you mean in this section because I know the details of GINA recommendations per year. However, any other reader will get confused. You have to re-write in a better way and cite the relevant reference in a logical consequence depending on the year and the rational for the change. The readers will prefer to go to the GINA to understand.
The mention of biomarkers in the title and text is not necessary because it overestimate what is introduced in this review. You can find recent (2022-2023) article about biomarkers, phenotypes, precision medicine in asthma, COPD, and their overlap (ACO). Also, consider the comparison of LAMA role in asthma (GINA 2022), COPD (GOLD 2023), and ACO (ACO_CP 2023).
The use of single triple therapy, of course, has an adherence advantage but with some disadvantages, like dose adjustments, treatment modifications etc, this make sense for all other treatments.
Needs modifications
Author Response
Reply to reviewer 3
First of all, we would like to thank you for your time and effort.
Reviewer
This review adds nothing or little to what is already available in the literature. It could be suitable as a letter after significant modifications.
Reviewer
Almost half of the references are old/very old (more than 5, 10, and even 20 years ago).
Response: Thank you for your point. However in Table 1 of the manuscript, that summarizes the main findings and limitations of the studies which were included in the present review, you can see that with the exception of 2 studies, all the other were published after 2018, which is less than 5 years ago.
Reviewer
Section 2.6:
This change in the GINA guideline was not in 2022 as you cited, but in 2019 version of GINA.
I understand what you mean in this section because I know the details of GINA recommendations per year. However, any other reader will get confused. You have to re-write in a better way and cite the relevant reference in a logical consequence depending on the year and the rational for the change. The readers will prefer to go to the GINA to understand.
Response: Thank you for your comment. We cited the latest GINA guidelines because they include the entirety of the recommendations for asthma management. In our opinion, it would be confusing to add every single GINA document as a reference based on the year of each recommendation, when the latest guidelines include all the recommendations. However, since it might be misleading, we deleted the word “latest” from the last sentence of the 1st paragraph of section 2.6 following your recommendation.
Reviewer
The mention of biomarkers in the title and text is not necessary because it overestimate what is introduced in this review. You can find recent (2022-2023) article about biomarkers, phenotypes, precision medicine in asthma, COPD, and their overlap (ACO). Also, consider the comparison of LAMA role in asthma (GINA 2022), COPD (GOLD 2023), and ACO (ACO_CP 2023).
Response: Thank you for your comment. We agree that the mention of biomarkers in the title might overestimate what is introduced in this review and we deleted it.
Reviewer
The use of single triple therapy, of course, has an adherence advantage but with some disadvantages, like dose adjustments, treatment modifications etc, this makes sense for all other treatments.
Response: Thank you for your comment. In accordance to GINA recommendations, the maintenance and reliever therapy with a single inhaler is the preferred treatment of choice compared to other alternative treatment choices. Given the effectiveness and the safety profile of LAMAs in the group of patients with severe uncontrolled asthma, although it is not supported by any relevant studies, at least for now, this review is the first to propose the combination of the preferable “Single Maintenance And Reliever Therapy” (SMART) with the “Single Inhaler Triple Therapy” (SITT) in a single device as the “optimal treatment strategy” for these patients. In this review, we have also quoted a figure (figure 1), in which we illustrate this treatment strategy so that it could be interpreted in an easier way. This is a novel suggestion which is NOT “already available in the literature” as you mentioned in your 1st comment. In our opinion, this is a very important aspect and future research should focus on that, in order to confirm or reject our hypothesis. After all, when maintenance and reliever therapy with a single inhaler was first introduced, not everybody expected it would become the preferred treatment choice according to GINA guidelines.
Reviewer
Comments on the Quality of English Language: Needs modifications
Response: Thank you for your comments
Round 2
Reviewer 3 Report
Some comments are addressed
Good
Author Response
Dear Reviewer,
Thank you again for the comment. We have deleted the mention of biomarkers in the title of the revised manuscript. We are sorry for the inconvenience.